

# Greater than *X* kb: a quantitative assessment of preservation conditions on genomic DNA quality, and a proposed standard for genome-quality DNA

Daniel G. Mulcahy[1], Kenneth S. Macdonald III[2], Seán G. Brady[3], Christopher Meyer[4], Katharine B. Barker[1] and Jonathan Coddington[1,3]

[1] Global Genome Initiative, National Museum of Natural History, Smithsonian Institution, Washington, DC, USA
[2] Laboratories of Analytical Biology, National Museum of Natural History, Smithsonian Institution, Washington, DC, USA
[3] Department of Entomology, National Museum of Natural History, Smithsonian Institution, Washingtion, DC, USA
[4] Department of Invertebrate Zoology, National Museum of Natural History, Smithsonian Institution, Washington, DC, USA

Corresponding author
Daniel G. Mulcahy,
MulcahyD@si.edu

## ABSTRACT

Advances in biodiversity genomic sequencing will increasingly depend on the availability of DNA samples—and their quantifiable metadata—preserved in large institutional biorepositories that are discoverable to the scientific community. Improvements in sequencing technology constantly provide longer reads, such that longer fragment length, higher molecular weight, and overall "genome-quality" DNA (gDNA) will be desirable. Ideally, biorepositories should publish numerical scale measurements of DNA quality useful to the user community. However, the most widely used technique to evaluate DNA quality, the classic agarose gel, has yet to be quantified. Here we propose a simple and economical method using open source image analysis software to make gDNA gel images quantifiable, and propose percentage of gDNA "greater than *X* kb" as a standard of comparison, where *X* is a band from any widely used DNA ladder with desirably large band sizes. We employ two metadata standards ("DNA Threshold" and "Percent above Threshold") introduced as part of the Global Genome Biodiversity Network (GGBN) Darwin Core extension. We illustrate the method using the traditionally used *Hin*dIII ladder and the 9,416 base-pair (bp) band as a standard. We also present data, for two taxa, a vertebrate (fish) and an invertebrate (crab), on how gDNA quality varies with seven tissue preservation methods, time since death, preservation method (i.e. buffers vs. cold temperatures), and storage temperature of various buffers over time. Our results suggest that putting tissue into a buffer prior to freezing may be better than directly into ultra-cold conditions.

## INTRODUCTION

Increasing the length of sequence reads is a core technological challenge in genomic science. Currently, the most widely used bench top technologies can achieve quality

reads up to 150–500 base-pairs (bp) in length (*Glenn, 2011*; *Loman et al., 2012*), but longer-read sequencing technologies are increasingly feasible (*Loman & Quinlan, 2014*). Therefore, access to high quality, high molecular weight DNA will become increasingly crucial. It is now feasible, practical, and increasingly more common to sequence complete genomes of non-model organisms (e.g. *GIGA Community of Scientists, 2014*; *Pisani et al., 2015*). As biodiversity genomics expands to rarer, harder-to-access, or vanishing organisms, obtaining "genomic quality" tissues—that provide high molecular weight DNA—becomes a significant challenge.

Natural history museums and academic institutions are currently obtaining, curating, and rapidly increasing biodiversity biobank collections (i.e. biorepositories), in order to maintain genomic quality material of non-model organisms, and to make this material available for scientific researchers conducting genomic analyses around the globe (*Droege et al., 2014*). In turn, making massive tissue and DNA collections discoverable is a priority for data aggregators, such as the Global Genome Biodiversity Network (GGBN; http://www.ggbn.org/ggbn_portal/). GGBN is a network of institutions dedicated to preserving genetic resources, but also to advancing the data model for tissues, DNAs, RNAs, and similar resources and their standardizations (*Droege et al., 2016*). As of August 2016, GGBN lists more than 500,000 samples of 32,000 species, 11,000 genera, and almost 2,200 families of life, located in 17 biobanks worldwide. 50 worldwide institutions are members of GGBN and are working to make their collections discoverable on the data portal, a 300% increase over the previous year.

We present a simple, cost-effective agarose gel electrophoresis method for qualitatively analyzing genomic DNA (gDNA) extractions (for genomic analyses) that can provide a quantifiable value of gDNA. Our method requires only basic molecular laboratory equipment [standard Tris-Borate-EDTA (TBE) gel rigs, UV imaging station, etc.]; thus it does not rely on any expensive reagents or more expensive analytical quantification equipment (e.g. spectrophotometers, automated electrophoresis systems, tape stations, pulsed field gel electrophoresis, etc.). Our aim is to suggest an inexpensive assay that biobanks could implement to indicate sample quality prospectively.

Studies that explore the impact of tissue preservation on DNA quality often measure DNA concentration or test whether particular loci will PCR-amplify and sequence using traditional Sanger methods (e.g. *Dawson, Raskoff & Jacobs, 1998*; *Vink et al., 2005*; *Yodder et al., 2006*; *Erkens et al., 2008*; *Frampton et al., 2008*; *Gaither et al., 2011*; *Moreau et al., 2013*). However, these approaches are limited, as even fragmented DNA may amplify and produce high quality Sanger sequencing products, particularly when these products are in the size-range for Sanger methods (e.g. ~500–1,000 bp). DNA concentration can be increased by adding more material (i.e. tissue) or combining multiple extractions from the same material, and does not provide any information about size. Genomic quality DNA should preferably be mostly intact (whole chromosomes and organelle genomes), particularly if the intention is to sequence entire genomes, as the assembly of degraded gDNA (non-randomly sheared) prior to library preparation can be problematic for most high-throughput sequencing (HTS) platforms (*Chen et al., 2015*). Therefore, in order to assess the quality of their DNA, many researchers use agarose

gels with high molecular-weight DNA ladders to visualize size and quality (*Williams, 2007*; *Gaither et al., 2011*), in addition to more sophisticated methods such as spectrophotometry, fluorometry, or automated electrophoresis methods.

Here, we propose a standardized, simple method for electrophoresing genomic DNA on agarose gels with the standard λ phage *Hin*dIII ladder. The size of the gDNA can be assessed by comparison to any of six bands in the *Hin*dIII ladder (2,027, 2,322, 4,361, 6,557, 9,416, and 23,130 bp), and from this comparison, the percent of gDNA greater than a given band size can be calculated from a regular gel image. We recommend this method to standardize quality assessment of tissues collected and reporting by biorepositories and data aggregators such as GGBN. We suggest use of the ~"9 kb" (= 9,416 bp) size marker as a working standard, because it is substantially longer than standard HTS reads (e.g. Illumina, etc.), and given current technologies, would be an appropriate minimum for long read sequencing. However, any size marker can be used to quantify gDNA, hence the method is referred to as "greater than *X* kb."

We present our method with a case study exploring preservative methods within field collection workflows that yield DNA of sufficient quality and quantity suitable for genomic sequencing. Within this case study, fresh, field collected tissue of a fish (*Morone americanus*) and a crab (*Callinectes sapidus*) were used to test if DNA quality is dependent on treatment and time until preservation. Alternative preservative solutions, temperature, and time were used as variables. Immediate cryopreservation was used as a benchmark for comparison as most researchers to date believe that freezing tissue at ultra-cold temperatures, such as −80 or −190 °C (liquid nitrogen), is the best preservative method for yielding genomic quality DNA.

We address four questions in this study: 1. Can DNA quality (in terms of fragment length) be measured quickly, consistently, and economically; 2. How does preservation method (buffers vs. temperature) affect DNA quality; 3. How does time since death affect DNA quality; and 4. How does storage temperature (in various buffers) affect DNA quality?

## MATERIALS AND METHODS
### Evaluating preservation methods

#### *Tissue collection*
Muscle tissue was collected from wild caught specimens of two species: *Morone americanus* (white perch, hereafter "fish") and *Callinectes sapidus* (blue crab, hereafter "crab") at the Smithsonian Environmental Research Center (SERC) in Edgewater, MD. The crabs were euthanized using liquid nitrogen asphyxiation (held above liquid nitrogen, which depletes oxygen), and the fish were euthanized with MS222, following our Animal Care and Use Committee (ACUC) protocols. Individual fish were filleted immediately after death. Fillets were cut into small strips (~0.5 × 5 cm) and immediately submerged in liquid nitrogen ($LN_2$). Individual crabs were dismembered upon death and claws were immediately submerged in $LN_2$. Two experiments were conducted ("Time" and "Temperature;" see below), and because this involved processing nearly 500 samples, all samples were submerged into $LN_2$ within five minutes after death to reduce postmortem

tissue degradation. Each experiment only used the tissue collected from a single individual (e.g., one fish for time, and one fish for temperature). For both experiments, each treatment combination consisted of ten replicates. Our study involving vertebrate animals was approved by the Smithsonian Institution, ACUC.

### Time experiment

Fillet strips and claws were thawed and sub-sampled; samples were weighed to the nearest mg before being subjected to one of seven preservation treatments: 1 ml 95% EtOH, 1 ml salt-saturated DMSO-EDTA buffer (modified from *Seutin, White & Boag, 1991*), 1 ml DNAzol (Molecular Research Center), 1 ml of RNAlater (Ambion), 300 µl M2 tissue digestion buffer (Autogen, Inc.), frozen at $-20\,°C$, frozen in $LN_2$ ($\approx -190\,°C$). The DMSO-EDTA buffer we use is a slight modification of the tissue buffer used by *Seutin, White & Boag (1991)*; we use 25% of DMSO, instead of 20% (with 25% of 0.5 M EDTA, 50% sterile $H_2O$, saturated with sodium chloride). Before being subjected to the preservation treatment, all samples were allowed to sit at room temperature (RT) for one of three time periods: preserved immediately after thawing (< 10 min total thaw time), 3 h after thawing, and 24 h after thawing. After 14 (crab) or 20 (fish) days, all sample tubes were moved into $LN_2$ and stored until DNA was extracted.

### Temperature experiment

Tissue samples were placed into one of five preservatives: 1 ml 95% EtOH, 1 ml DMSO-EDTA, 1 ml DNAzol, 1 ml RNAlater, 300 µl M2 lysis buffer and then stored for 15 (crab) to 21 (fish) days at one of five temperatures: RT, 4, $-20$, $-80$, and $-190\,°C$ ($LN_2$). Subsequently, all sample tubes were stored in liquid nitrogen until DNA was extracted.

### DNA extraction

All samples were digested overnight in 300 µl AutoGen M2 and 300 µl M1 buffer (including Proteinase K). DNA was extracted from 300 µl (½ of the digested amount) of each digested sample by an AutoGen Prep 965 automated DNA extractor (AutoGen Inc., Holliston, MA, USA) using the manufacturers standard animal tissue (phenol-chloroform) extraction method, and then dried. Samples were eluted in either 100 µl (fish) or 50 µl (crab) R9 DNA re-suspension solution (AutoGen Inc., Holliston, MA, USA).

### DNA quantification

All sample extractions were quantified through fluorescence, using a BioTek Synergy HT Multi-Mode Microplate Reader and Quant-iT dsDNA Assay Kit, broad range (Invitrogen, Cat# Q33130). The 8 µl of eluted DNA was added to 200 µl of buffer and 1 µl of reagent in an opaque black 96-well microplate (Corning, Cat# 3915) and mixed thoroughly. The 10 µl of seven solutions with known dsDNA concentrations (0, 10, 20, 40, 60, 80, and 100 ng/µl) were each added to two wells on every plate to calculate standard concentration curves. After sitting at RT for three minutes, samples were excited at 485 nm and ensuing fluorescence was read at 528 nm. Each sample was read twice, with five minutes between reads, and reads were averaged. Duplicate fluorescence values from

concentration standards were averaged, and a general linear model of these fluorescence values versus total known DNA amounts was calculated using the program R, with the intercept constrained to run through the origin. This model was then used to calculate the total ng of DNA in each sample. The DNA concentration of the extraction was calculated by dividing total DNA by eight (the volume of sample used). We also calculated the total DNA extracted by multiplying fluorescent sample total DNA by 12.5 (for fish, because it was eluted in 100 $\mu$l) or 6.25 (for crab, because it was eluted in 50 $\mu$l). Finally, we calculated a DNA extraction yield (ng DNA/mg Tissue) by dividing the total DNA extracted by the weight of each tissue sample, and multiplying by two (because only half of each digest was used in the extraction).

### Statistical analyses

Treatment differences were evaluated separately for each species (fish, crab) and experiment (Time, Temperature). The program R was used to run two-factor Analyses of Covariance (ANCOVA) on the quality and quantity datasets separately. Analyses included a time or temperature by preservative interaction term and used tissue weight as a covariate. If weight was non-significant, it was removed and the same ANCOVA, but without the covariate, was run. In total, 14 analyses were conducted for each experiment. Therefore, a Bonferroni correction of $\alpha = 0.004$ was used as a measure of significance.

## Gel quantification: greater than *X* kb

### Gel electrophoresis protocol

Extracted gDNA for all samples was visualized on a 1% agarose TBE gel. The 5 $\mu$l of each gDNA extract was loaded into the gel, and electrophoresed at 45 v for 2.5 h in 1X TBE buffer. To estimate gDNA fragment length, 0.5–1 ug of *Hin*dIII ladder was loaded into wells on each side of the DNA-loaded wells. After electrophoresis, gels were stained for 30 min in a solution of Ethidium Bromide (EtBr; at a final concentration of 0.5 $\mu$g/ml) and 1X TBE buffer. Gels were subsequently de-stained (to reduce background staining) in H$_2$O for 15 min. Finally, gDNA was visualized, photographed and images were stored as TIFF files using a Syngene Gene GeneGenius Bio Imaging System.

Additionally, because EtBr is carcinogenic, and many labs are moving away from its use and replacing it with safer methods, we also optimized the gel electrophoresis protocol with GelRed™ (BioTium, Fremont, CA, USA). For this method, we recommend a 0.7% agarose TBE gel, run for ~2.5 h at 45 v in 1X TBE buffer. When using GelRed™, we recommend loading a few wells of diluted *Hin*dIII ladder in various amounts (e.g. 1:24, 1:49, and 1:99) because the GelRed™ can cause wide smearing in the *Hin*dIII ladder bands (Fig. 1). For either staining method, it is important to have *Hin*dIII ladder on either ends of the gDNA, such that it "brackets" the gDNA samples on either end for post-scoring of the gel images (Fig. 1).

### Terminology

The GGBN has created a set of vocabularies, referred to as the "Data Standard," designed to represent genomic (tissue, DNA, etc.) samples associated with voucher specimens, complementing the Access to Biological Collection Data (ABCD) and Darwin Core

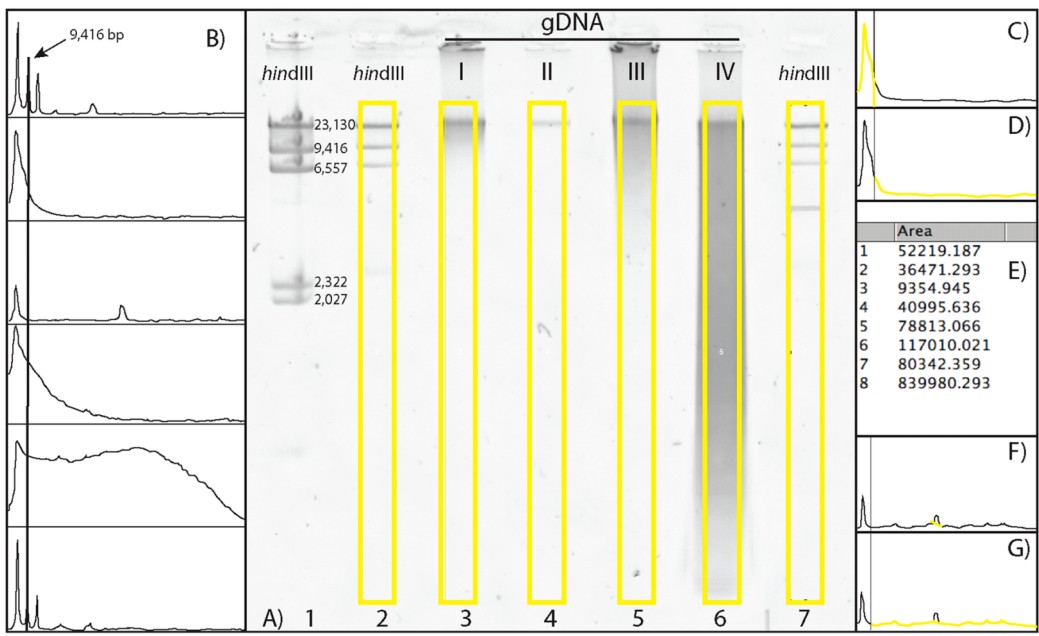

**Figure 1 Scoring a gDNA gel in ImageJ.** (A) An inverted gDNA gel image with the *Hin*dIII ladder in wells 1–2 (wells labeled on bottom) and 7, and gDNA samples (I–IV) in wells 3–6. The bands in the *Hin*dIII ladder are labeled in well 1; note the ~2 kb bands are visible in this lane, but not in the more diluted lanes (2 and 7; see text). (B) The Plot results of the six yellow boxes scored in A), with the Straight tool line spanning the ~9 kb peaks of the two *Hin*dIII ladders (wells 2 and 7, top and bottom, respectively), with the four gDNA samples in the middle. (C) The area calculated for > 9 kb. (D) The area < 9 kb. (E) The Results box, with values 1 and 2 for the areas > 9 and < 9 kb for gDNA sample I, respectively. (F) The Straight tool line used to eliminate peak cause by flaw in gel for gDNA sample II. (G) The calculated area < 9 kb with the erroneous peak removed. The percent of gDNA > 9 kb is calculated by dividing the area > 9 kb (C) by the total area below the curve (C + D).

standards (*Droege et al., 2016*). In this list is a set of GGBN Gel Image Vocabulary available at: http://terms.tdwg.org/wiki/GGBN_Gel_Image_Vocabulary, where all terms are defined. Two of the most relevant terms here are defined as follows: The DNA Threshold (http://terms.tdwg.org/wiki/ggbn:DNAThreshold): Fragment size of the ladder chosen as a standard to measure the percentage of DNA size, at or above threshold fragment size from ladder; Percent Above Threshold (http://terms.tdwg.org/wiki/ggbn:percentAboveThreshold): Percent of DNA at or above the size of the threshold.

### Scoring of the gels

Specific, step-by-step instructions can be found on the online *Supplementary Information Appendix I*.

To visualize and score the gel images, we used the program ImageJ v1.48 (W. Rasband, NIH). ImageJ is an open-source, Java based program in the public domain, available at http://rsb.info.nih.gov/ij. Gel image files are opened in ImageJ, with the wells of the gel at the top. The gel image is color inverted to enhance visualization of the bands and make density curves positive, rather than negative (see below). The image is made level with respect to the top band in each ladder on either side of the gDNA samples, the background is subtracted to remove smooth continuous backgrounds from the
gel images. Next, a vertical box is drawn encompassing the entire length of the lane containing *Hin*dIII ladder, starting below the well, above the ∼23 kb band, and to the extent of the gDNA (Fig. 1A). Vertically, the box should encompass the entire length of the smallest fragments of the gDNA lane on the gel with the greatest range.

The *Hin*dIII ladder box on the left is selected as the first lane. If using multiple dilutions of *Hin*dIII ladder, select the dilution that has the clearest bands on both sides of your gDNA samples. The box over the *Hin*dIII ladder is then moved to the first gDNA lane, and this lane is selected as the next to be analyzed (note: when moved, the original box stays in place and a new one is placed over the next lane). The horizontal location of this box must be carefully selected, so that the bands (or smears) of gDNA encompass the entire width of the box. The program automatically adjusts the vertical placement to be level with the first box. Then, the box from the first gDNA lane is moved to the next gDNA lane to be scored (typically, the next one to the right), again carefully selecting the location of the new lane horizontally, centering the box on the gDNA. Additional gDNA lanes are selected from left to right, by dragging the previous box from the left to the right, and selecting "Next Lane," until all desired lanes are included. The last lane selected must be the *Hin*dIII ladder to the right of gDNA lanes (Fig. 1A). An "Intensity Plot" is then created of the selected lanes, including the *Hin*dIII ladder lanes.

The Intensity Plot opens in a new window and is rotated 90° clockwise from the gel orientation (i.e. the leftmost lane becomes the top intensity plot). A straight line is drawn for the DNA Threshold (Fig. 1B) from the apex of the ladder threshold peak (e.g. "9 kb") on the first ladder (top of the Intensity Plot) to the apex of the ladder threshold peak on the second ladder (bottom of the Intensity Plot). This separates the intensity curves of each lane into a region greater than the threshold peak (Fig. 1C) and a region less than the size of the threshold peak (Fig. 1D). All regions must be closed to be measurable; if the right side of the intensity curve does not meet the vertical line at the right side of the plot (leaving this region open), the Straight tool is used to draw a vertical line connecting the right part of the curve to the border of the plot.

The Wand tool is then used to select and measure the area of a region under the Intensity Plot, on the left side of the vertical line (> 9 kb) and to the right of the vertical line (< 9 kb). Once selected, the area is automatically calculated and presented in a Results table (Fig. 1E). If a dark imperfection appears in the gel that is clearly not part of the gDNA (Fig. 1A, gDNA II), a peak is recorded in the Intensity Plot. Similar to closing areas to measure (described above), one can eliminate the erroneous peak by using the Straight tool (Fig. 1F), and recalculate the area (Fig. 1G). The numbers in the Results window are the areas of the curve greater and less than the size of the threshold peak, respectively (Fig. 1E). These data are then copied and pasted into a data processing file and the Percent Above Threshold (e.g. ∼9 kb) is easily calculated by dividing the area to the left (> 9 kb), by the total area.

### Scoring tests

We ran two analyses to test the repeatability (the variation obtained when one person measures samples repeatedly using the same methods) and reproducibility (the variation

obtained when multiple people measure samples repeatedly using the same methods) of our gel scoring method. In the first analysis, two co-authors (CM and KSM) each independently scored the same gel image (FishTime < 10 min) consisting of 40 lanes of gDNA, 10 times. For each scoring, the entire process was repeated, starting with opening the raw image in ImageJ. Additionally, each scoring process was timed to give an estimate of method efficiency. The results of the 20 scored gel images (consisting of 800 scored gDNA lanes) were analyzed using a Gage repeatability and reproducibility ANOVA (Gage R&R) using the spreadsheet devised by J. Muelaner (http://www.muelaner.com/quality-assurance/gage-r-and-r-excel/). In order to test the consistency of dDNA in the gels, for the second analysis we ran seven gDNA samples, four crab (from Time < 10 min) and three fish (from Temp = 4 °C), each multiple times on two different gels. Each crab sample was run three times on each gel, while each fish sample was run four times on each gel. The gel images were also independently scored by two co-authors (DM and KSM), and the results of this test were also analyzed using a Gage R&R.

## RESULTS

### Time experiment

Figure 2A shows the fish gDNA extractions run out on a gel with the *Hin*dIII ladder from the seven preservation methods at time < 10 min after death. The DMSO-EDTA and DNAzol buffers have the greatest percent of gDNA > 9 kb (72 and 87%, respectively), consistent with the gel patterns showing the largest bands of gDNA, with little streaking or smearing in the lanes, indicating very little fragmented DNA. Figure 2B shows the results of the fish gDNA extractions from samples preserved at RT in 95% EtOH at < 10 min, 3, and 24 h after death before being frozen in $LN_2$. The gDNA degrades through time resulting in little to no high molecular weight DNA after 24 h (Fig. 2B).

Figure 3 shows the gel image of the crab gDNA for time < 10 min, for seven different preservation methods. The EtOH, DMSO, and DNAzol gDNAs have the greatest percentage of gDNA > 9 kb (93, and 100%, respectively), and show less smearing and larger fragment size than RNAlater, −20, −190 °C, and M2.

Average measures of quantities, concentrations, and quality (as measured by % of DNA > 9,416 bp) are shown for each trial for the time experiment for the fish and crab gDNA extractions in Table 1. Figures 4 and 5 show the quality of fish and crab gDNA (% > 9 kb), respectively, for seven different preservation methods over three time periods. DNA quality varies greatly in both taxa at time < 10 min, but all methods show degradation in quality of gDNA over time for both fish and crab tissues.

### Temperature experiment

Figure 6A shows the fish gDNA extractions electrophoresed on a gel with the *Hin*dIII ladder from five different preservation methods at RT. Note the DMSO-EDTA, DNAzol, and RNAlater buffers have the greatest percentage of gDNA > 9 kb (89, 98, and 100%, respectively), and show the largest bands of gDNA with little streaking or smearing in the lanes, indicating very little fragmented gDNA. Figure 6B shows the results of the fish

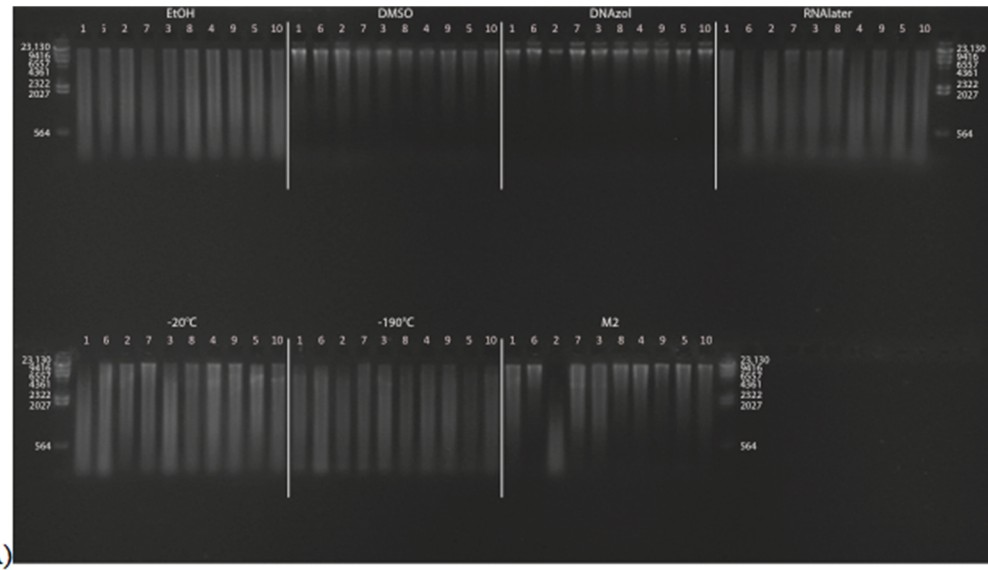

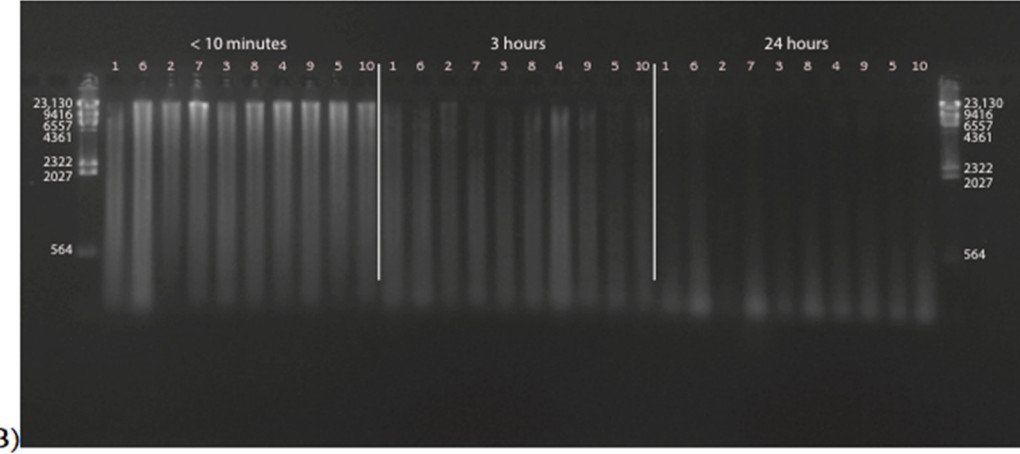

**Figure 2 Gel image of extracted gDNA from the white perch, *Morone americana* for the Time Experiment.** The ladder in the left- and right-most lanes is the *Hin*dIII with bands at 564, 2,027, 2,322, 4,361, 6,557, 9,416, and 23,130 bp. (A) Samples in all tissue storage treatments for Time-since-death < 10 min: 1) in EtOH; 2) in salt saturated DMSO/EDTA preservation buffer ("DMSO"); 3) submerged in DNAzol Reagent (Invitrogen); 4) submerged in RNAlater (Ambion); 5) held at −20 °C with no preservation solution; 6) submerged in liquid nitrogen (≈ −190 °C) with no preservation solution; 7) submerged in M2 tissue digestion solution (Autogen). (B) Samples in Time-since-death treatments for EtOH tissue storage treatment are shown for the three different time periods (< 10 min, 3, 24 h).

gDNA extractions electrophoresed on a gel (with the *Hin*dIII ladder) from the five different temperatures the tissue were stored at while in the DMSO-EDTA buffer. Figure 7 shows the gel image of the crab gDNA for the different preservation methods stored at RT. Average measures of quantities, concentrations, and quality (as measured by % of DNA > 9,416 bp) are shown for each trial for the temperature experiment for the fish and crab gDNA extractions in Table 2. Figures 8 and 9 show the quality of fish and crab gDNA (% > 9 kb), respectively, for the different preservation methods over the five temperatures at which tissues were stored.

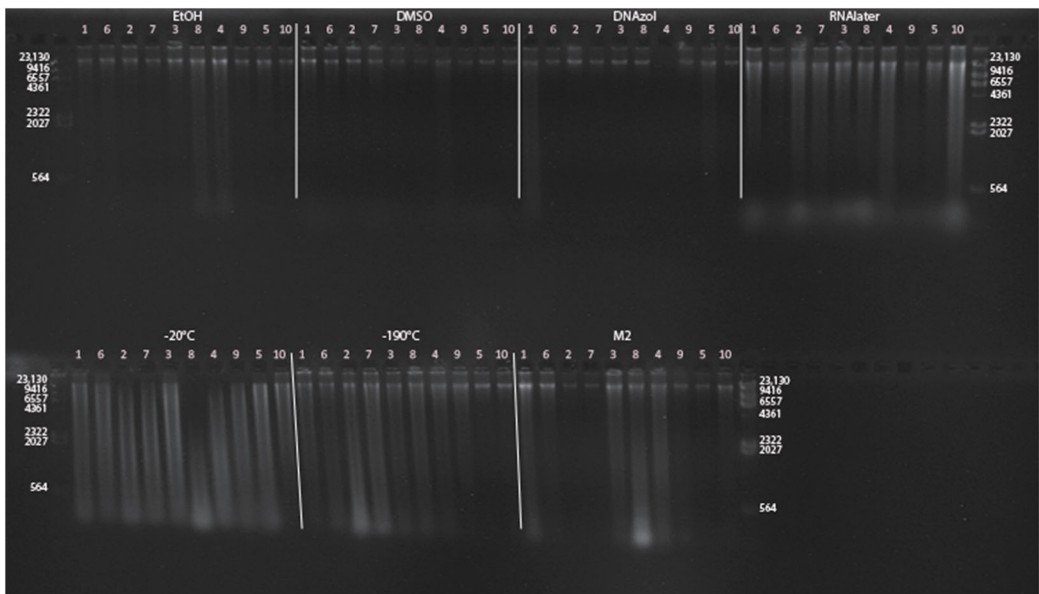

**Figure 3 Gel image of extracted genomic DNA from the blue crab, *Callinectes sapidus*, for the Time Experiment, showing all tissue storage treatments for Time-since-death < 10 min.** The *Hin*dIII ladder is shown in the left- and right-most lanes of the gel.

## ANCOVA statistics

Table 3 shows the results of the analysis of covariance statistics for the fish and crab, Time and Temperature experiments, for both quality (% of gDNA > 9 kb) and quantity (ng gDNA/mg tissue) of gDNA. In the fish Time Experiment, time, preservation method, and the interaction variable all significantly affected DNA quality, while only time had a significant affect on quantity. For both quality and quantity, the covariate weight was marginally insignificant. In the crab Time Experiment, all factors (time, preservative, time × preservative, weight) significantly affected gDNA quality. The interactive term was significant only for gDNA quantity, although preservation method was just marginally insignificant. For the fish Temperature Experiment, only preservation method significantly affected gDNA quality, while temp, preservation method and weight all affected quantity. Finally, in the crab Temperature Experiment, only preservation method significantly affected either quality or quantity, although weight was marginally insignificant for both (Table 3).

## Repeatability and reproducibility

The first (single gel) test resulted in the repeatability variation (the % of total variation that is attributable to the scorer, i.e. the variability among the 10 scores a single co-author gave the same sample) of 3.99% and reproducibility variation (the % of total variation attributable to differences in the way the co-authors scored the same sample) of 5.71%. The total Gage Repeatability and Reproducibility variation was 6.97%. The second (multiple gel) test had a repeatability variation of 22%, a reproducibility variation of 18.1%, and a total Gage R&R variation of 28.5%.

**Table 1 Time experiment.** Average quantities of Extracted DNA (ng), DNA extraction yield (ng DNA/mg tissue digested; see text for method of calculation) and Quality (% DNA > 9,416 bp) of genomic DNA extracted from tissues of two species of fish (*Morone americana*) and crab (*Callinectes sapidus*) held in one of 21 treatments: three times prior to preservation (< 10 min, 3, and 24 h), then stored in seven preservative methods (95% EtOH, DMSO-EDTA, DNAzol, RNAlater, M2, −20, and −190 °C) for a minimum of 14 days. Mean and standard deviation (SD) values are shown.

| | < 10 min | | | | | | 3 h | | | | | | 24 h | | | | | |
|---|---|---|---|---|---|---|---|---|---|---|---|---|---|---|---|---|---|---|
| | Extracted DNA (ng) | | Extraction yield (ng DNA/mg tissue) | | Quality (% > 9,416 bp) | | Extracted DNA (ng) | | Extraction yield (ng DNA/mg tissue) | | Quality (% > 9,416 bp) | | Extracted DNA (ng) | | Extraction yield (ng DNA/mg tissue) | | Quality (% > 9,416 bp) | |
| | Mean | SD | Mean | SD | Mean | SD | Mean | SD | Mean | SD | Mean | SD | Mean | SD | Mean | SD | Mean | SD |
| Fish: *Morone americana* | | | | | | | | | | | | | | | | | | |
| EtOH | 186 | 28 | 78 | 10 | 6 | 3 | 209 | 49 | 83 | 18 | 4 | 4 | 98 | 14 | 40 | 7 | 1 | 2 |
| DMSO | 183 | 57 | 81 | 20 | 72 | 10 | 171 | 94 | 71 | 40 | 25 | 8 | 13 | 5 | 6 | 2 | 0 | 0 |
| DNAzol | 244 | 170 | 108 | 71 | 87 | 5 | 157 | 25 | 63 | 12 | 34 | 8 | 14 | 4 | 6 | 2 | 0 | 0 |
| RNAlater | 182 | 17 | 81 | 8 | 7 | 6 | 180 | 36 | 72 | 20 | 0 | 0 | 86 | 37 | 38 | 18 | 0 | 0 |
| M2 | 214 | 44 | 101 | 21 | 35 | 20 | 174 | 31 | 79 | 13 | 4 | 4 | 84 | 18 | 40 | 8 | 3 | 5 |
| −20 °C | 236 | 37 | 102 | 15 | 7 | 5 | 166 | 29 | 69 | 11 | 1 | 2 | 110 | 45 | 49 | 19 | 2 | 5 |
| −190 °C | 191 | 44 | 84 | 15 | 3 | 2 | 165 | 24 | 70 | 10 | 0 | 2 | 137 | 43 | 59 | 18 | 0 | 0 |
| Crab: *Callinectes sapidus* | | | | | | | | | | | | | | | | | | |
| EtOH | 97 | 48 | 22 | 10 | 93 | 5 | 369 | 312 | 79 | 64 | 46 | 36 | 303 | 94 | 61 | 17 | 1 | 1 |
| DMSO | 38 | 11 | 8 | 3 | 93 | 13 | 136 | 85 | 30 | 15 | 48 | 22 | 28 | 15 | 5 | 2 | 0 | 0 |
| DNAzol | 34 | 28 | 8 | 7 | 100 | 0 | 130 | 108 | 28 | 23 | 20 | 10 | 14 | 4 | 3 | 1 | 0 | 0 |
| RNAlater | 140 | 74 | 28 | 14 | 81 | 17 | 231 | 126 | 52 | 30 | 53 | 37 | 184 | 33 | 35 | 7 | 0 | 0 |
| M2 | 119 | 88 | 34 | 26 | 75 | 18 | 250 | 114 | 64 | 29 | 56 | 32 | 31 | 40 | 8 | 10 | 0 | 0 |
| −20 °C | 275 | 56 | 58 | 12 | 17 | 16 | 278 | 143 | 70 | 43 | 26 | 37 | 196 | 66 | 43 | 7 | 0 | 0 |
| −190 °C | 209 | 112 | 44 | 23 | 72 | 16 | 381 | 214 | 90 | 47 | 49 | 36 | 102 | 141 | 19 | 26 | 0 | 0 |

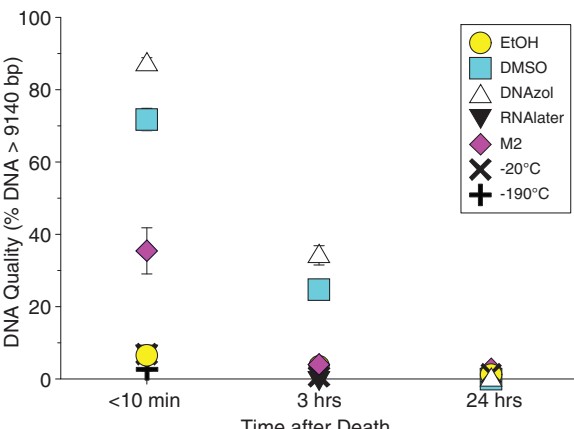

**Figure 4 Graph of fish Time Experiment.** Quality of gDNA extracted from the white perch, *Morone americana*. Quality is shown as the mean % of gDNA > 9,416 bp (±SE) on the y-axis. Preservation treatments are differentiated by symbols (see legend). Time-since-death treatments are shown on the x-axis for the three time periods tissue samples sat at room temperature before preservation. See Table 1 for exact values of each method.

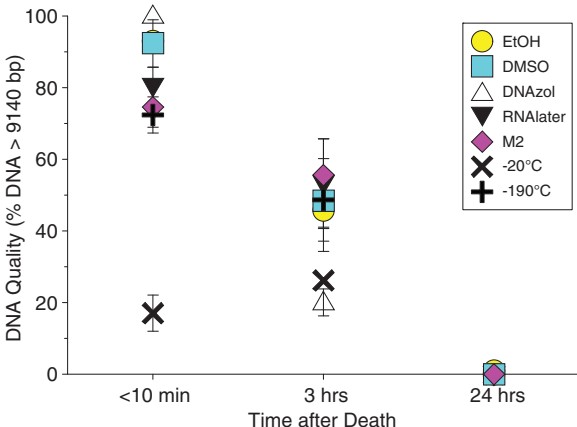

**Figure 5 Graph of crab Time Experiment.** Quality of gDNA extracted from the blue crab, *Callinectes sapidus*. Quality of extracted gDNA is shown as the mean % of DNA > 9,416 bp (±SE) on the y-axis. Preservation treatments are differentiated by symbols (see legend). Time-since-death differentiated are shown on the x-axis for the three time periods tissue samples sat at room temperature before preservation. See Table 1 for exact values of each method.

# DISCUSSION

## Greater than *X* kb

Here we demonstrate a simple, consistent, and efficient method for determining the size and quality of genomic DNA that does not require expensive equipment or reagents. Previous studies have presented the effects of different preservation conditions on DNA without providing an objective metric for genomic quality as we have done (e.g., *Gaither et al., 2011*; *Camacho-Sanchez et al., 2013*). We propose this method as a heuristic standard for biodiversity biobanking facilities and the genomic community, which may desire an inexpensive, approximate assessment of DNA quality before requesting tissue samples.

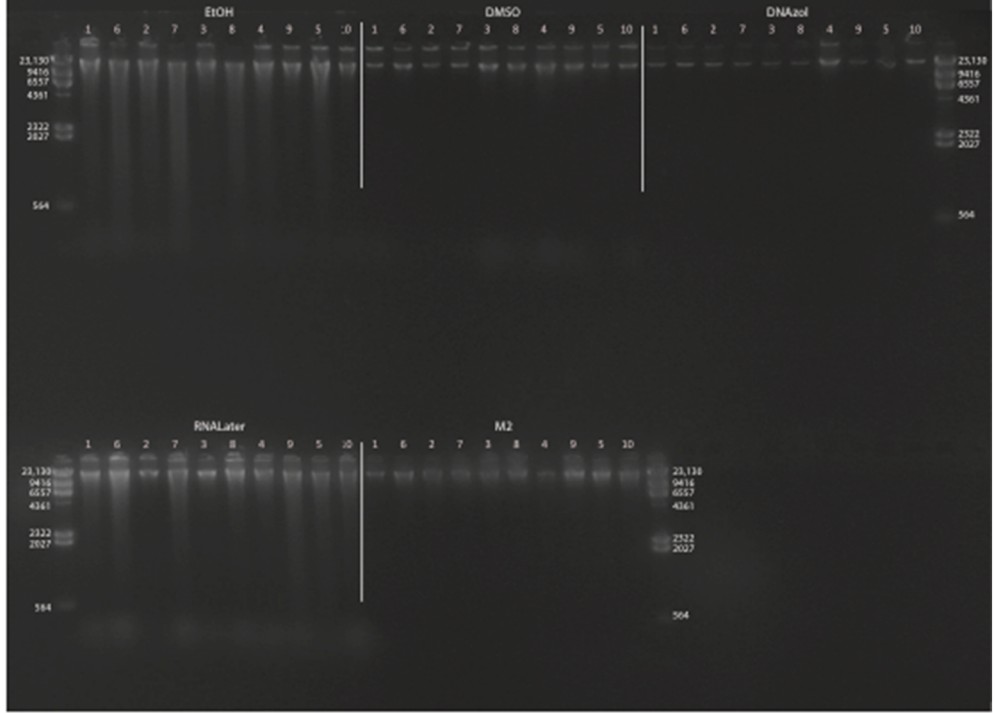

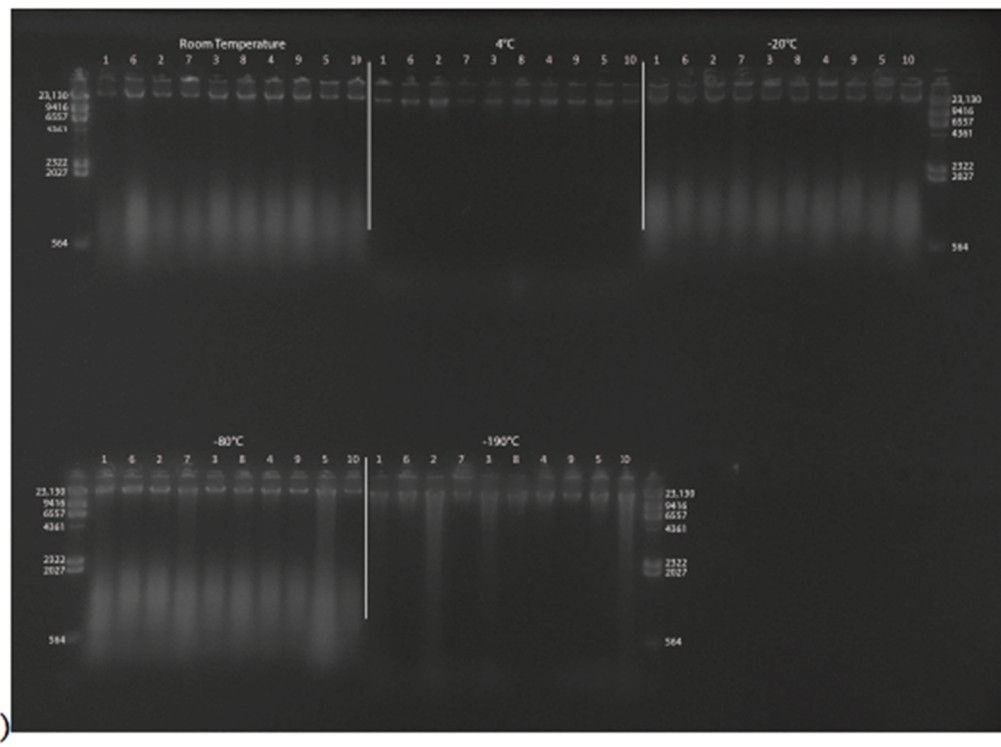

**Figure 6  Gel image of extracted gDNA for the white perch,** *Morone americana,* **for the Temperature Experiment.** Prior to DNA extraction, tissue was stored for 14–20 days in one of five solutions (EtOH, DMSO, DNAzol, RNAlater, M2) and kept at one of five temperatures: Room Temperature, −20, −80, and −190 °C. (A) Showing all tissue storage buffer treatments for tissue storage at Room Temperature. (B) Showing all tissue storage temperature treatments for DMSO-EDTA salt buffer.

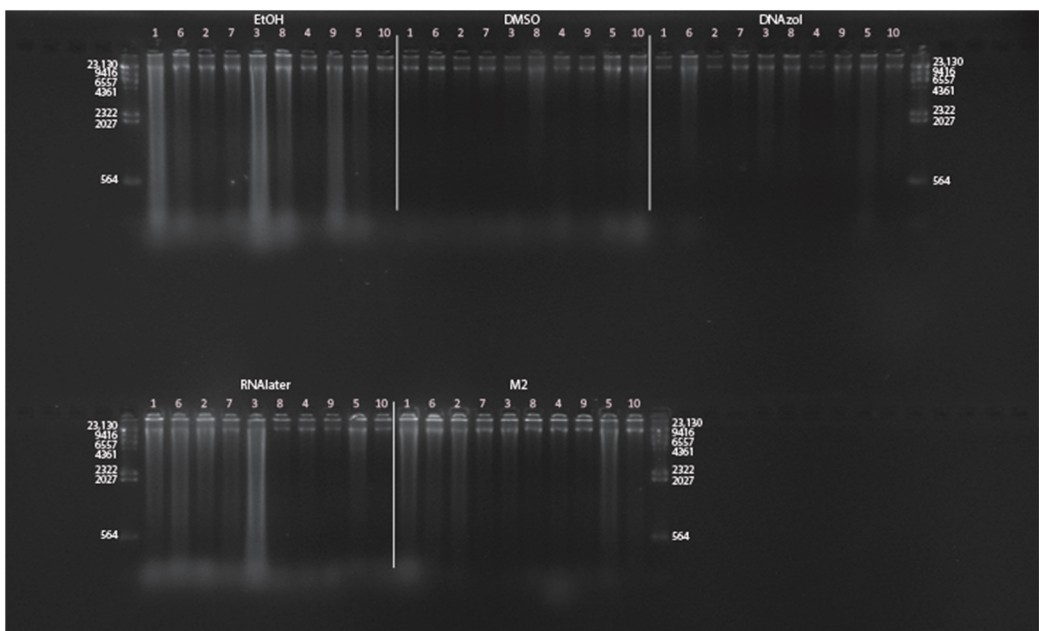

**Figure 7 Gel image of extracted genomic DNA for the blue crab, *Callinectes sapidus* for preservation Temperature Experiment.** All tissue storage buffer treatments for tissue storage at room temperature are shown.

Considering that that largest publicly accessible database of genomic samples already totals 500,000 and is growing rapidly (*Droege et al., 2016*), a rapid and cheap assay could be a useful heuristic tool. Using this method, genomic DNA can be electrophoresed on an agarose gel with a *Hin*dIII ladder, or any other large-sized DNA marker, and quantified using simple plots in the free software ImageJ (W. Rasband, NIH: http://rsb. info.nih.gov/ij).

We tested whether single or multiple researchers could reliably score a gel image similarly multiple times using a using a Gage Repeatability and Reproducibility ANOVA. The Gage R&R evaluates repeatability by the amount of variation attributable to a single measurer through multiple measures of the same sample, and evaluates reproducibility by the amount of variation attributable to differences between measurers. The analysis also calculates an overall measure of repeatability and reproducibility (the total Gage R&R), where most guidelines consider any total Gage R&R values under 10% to be acceptable, and any total values under 30% to be acceptable under certain conditions (*Pan, 2006*). For this test, the repeatability variation was 3.99%, the reproducibility variation was 5.7%, and the Total Gage R&R was 6.97%. Both researchers gave the same sample image similar quality scores (though not surprisingly, quality scores were slightly more different between researchers than were scores by the same researcher) to an extent acceptable to most quality control applications.

We conducted a second test to evaluate quality scores from samples run multiple times on a gel, and on separate gels. Our Gage R&R variability was much higher for this experiment, with a repeatability variation of 22%, a reproducibility of 18%, and a Total Gage R&R variation of 28.5%. There are many factors that can affect these scores in

**Table 2 Temperature experiment.** Average quantities of Extracted DNA (ng), DNA extraction yield (ng DNA/mg tissue digested, see text for method of calculation) and Quality (% DNA > 9,416 bp) of genomic DNA extracted from tissues of two species fish (*Morone americana*) and crab (*Callinectes sapidus*) held in one of 25 treatments: five storage temperatures (Room Temperature = RT, 4, −20, −80, and −190 °C) × five preservative methods (95% EtOH, DMSO-EDTA, DNAzol, RNAlater, M2) for a minimum of 14 days. Mean and standard deviation (SD) values are shown.

| | RT | | | | | | 4 °C | | | | | | −20 °C | | | | | | −80 °C | | | | | | −190 °C | | | | | |
|---|---|---|---|---|---|---|---|---|---|---|---|---|---|---|---|---|---|---|---|---|---|---|---|---|---|---|---|---|---|---|
| | Extracted DNA (ng) | | Extraction yield (ng DNA/mg tissue) | | Quality (% > 9,416 bp) | | Extracted DNA (ng) | | Extraction yield (ng DNA/mg tissue) | | Quality (% > 9,416 bp) | | Extracted DNA (ng) | | Extraction yield (ng DNA/mg tissue) | | Quality (% > 9,416 bp) | | Extracted DNA (ng) | | Extraction yield (ng DNA/mg tissue) | | Quality (% > 9,416 bp) | | Extracted DNA (ng) | | Extraction yield (ng DNA/mg tissue) | | Quality (% > 9,416 bp) | |
| | Mean | SD | Mean | SD | Mean | SD | Mean | SD | Mean | SD | Mean | SD | Mean | SD | Mean | SD | Mean | SD | Mean | SD | Mean | SD | Mean | SD | Mean | SD | Mean | SD | Mean | SD |
| **Fish: *Morone americana*** | | | | | | | | | | | | | | | | | | | | | | | | | | | | | | |
| EtOH | 233 | 129 | 150 | 69 | 46 | 18 | 229 | 87 | 157 | 54 | 23 | 12 | 188 | 38 | 120 | 23 | 41 | 11 | 159 | 31 | 105 | 17 | 49 | 24 | 306 | 105 | 209 | 71 | 32 | 17 |
| DMSO | 375 | 170 | 248 | 96 | 89 | 10 | 295 | 116 | 196 | 66 | 89 | 8 | 309 | 155 | 208 | 108 | 94 | 6 | 372 | 192 | 263 | 132 | 95 | 13 | 488 | 244 | 326 | 189 | 82 | 20 |
| DNAzol | 298 | 177 | 194 | 109 | 98 | 4 | 297 | 114 | 192 | 74 | 83 | 15 | 294 | 143 | 202 | 90 | 96 | 6 | 447 | 184 | 307 | 127 | 100 | 0 | 609 | 238 | 403 | 157 | 40 | 42 |
| RNAlater | 173 | 127 | 125 | 102 | 100 | 0 | 169 | 44 | 109 | 23 | 59 | 14 | 160 | 28 | 111 | 13 | 81 | 6 | 184 | 78 | 125 | 57 | 77 | 10 | 174 | 61 | 117 | 35 | 71 | 23 |
| M2 | 161 | 73 | 110 | 52 | 82 | 4 | 180 | 33 | 134 | 18 | 30 | 17 | 188 | 87 | 142 | 64 | 80 | 10 | 164 | 45 | 116 | 31 | 88 | 6 | 269 | 66 | 172 | 33 | 63 | 13 |
| **Crab: *Callinectes sapidus*** | | | | | | | | | | | | | | | | | | | | | | | | | | | | | | |
| EtOH | 1,238 | 354 | 430 | 112 | 52 | 35 | 889 | 190 | 329 | 81 | 62 | 24 | 804 | 432 | 291 | 159 | 39 | 14 | 1,264 | 580 | 415 | 173 | 24 | 4 | 942 | 222 | 320 | 87 | 77 | 24 |
| DMSO | 888 | 367 | 320 | 155 | 79 | 6 | 388 | 162 | 141 | 54 | 37 | 20 | 727 | 345 | 257 | 98 | 45 | 27 | 864 | 472 | 308 | 186 | 24 | 3 | 715 | 284 | 241 | 101 | 49 | 28 |
| DNAzol | 643 | 252 | 242 | 100 | 90 | 10 | 705 | 436 | 257 | 177 | 77 | 27 | 511 | 371 | 171 | 110 | NA | NA | 920 | 270 | 332 | 110 | 28 | 3 | 432 | 213 | 151 | 72 | 91 | 12 |
| RNAlater | 641 | 175 | 229 | 56 | 62 | 23 | 704 | 242 | 271 | 108 | 55 | 11 | 399 | 131 | 142 | 50 | 54 | 7 | 772 | 667 | 289 | 271 | 36 | 16 | 532 | 320 | 178 | 90 | 65 | 24 |
| M2 | 582 | 166 | 222 | 67 | 84 | 15 | 606 | 282 | 227 | 105 | 73 | 21 | 750 | 194 | 268 | 81 | 59 | 19 | 848 | 474 | 309 | 171 | 78 | 23 | 1,024 | 273 | 336 | 93 | 60 | 28 |

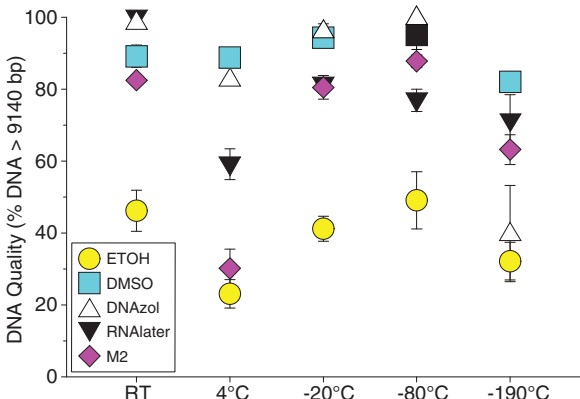

**Figure 8 Graph of fish Temperature Experiment.** Quality of gDNA extracted from white perch, *Morone americana*. Quality of extracted gDNA is shown as the mean % of gDNA > 9,416 bp (±SE) on y-axis. Preservation solutions are differentiated by symbols (see legend). Temperatures are shown on the x-axis for the five temperatures tissue samples were held after preservation. See Table 2 for exact values of each temperature.

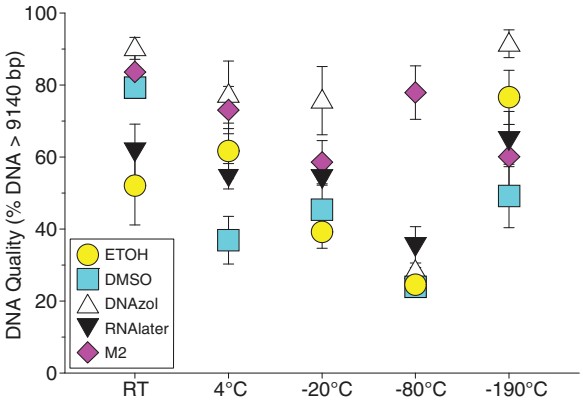

**Figure 9 Graph of crab Temperature Experiment.** Quality of gDNA extracted from the blue crab, *Callinectes sapidus*. Quality of extracted gDNA is shown as the mean % of gDNA > 9,416 bp (±SE) on the y-axis. Preservation treatments are differentiated by symbols (see legend). Temperatures are shown on the x-axis for the five temperatures tissue samples were held after preservation. See Table 2 for exact values of each temperature.

addition to measurer variability, such as variation in image quality (camera exposure, focus, dynamic range), variation in gel staining (length of time, mixing of stain, type of stain), and pipetting variation. Minor nuances between gel runs can result in slight discrepancies of quality scores, but in our case most values of the same sample were within ~5% of individual scores.

For the purposes of demonstrating the method, we chose the 9,416 bp (~"9 kb") size marker from *Hin*dIII as the standard, and reported the percentage of genomic DNA greater than 9 kb, with the recognition of 50% or more of the gDNA being greater than 9 kb as a candidate indicator of "genomic quality." Although other fragment sizes could be chosen, in our experience with legacy biorepository samples, many gDNA extractions will fail to meet a higher standard. HTS techniques show promising capabilities of producing reads much longer than 9 kb (*Loman & Quinlan, 2014*);

Table 3 **Summary of ANCOVA results for time and temperature experiments.** Covariance was tested for between Time, Preservation method (Pres.), and Weight (Wt.) for the Time experiments and Temperature (Temp.), Preservation method (Pres.), and Weight (Wt.) for the Temperature experiments. Interactions (Int.) were tested for between Time and Temp. and Pres., if weight was not significant it was removed. A Bonferroni correction of α = 0.004 is used.

| | | Time experiment | | | | Temperature experiment | | | |
|---|---|---|---|---|---|---|---|---|---|
| | | Time | Pres. | Wt. | Int. | Temp. | Pres. | Wt. | Int. |
| **Fish:** | | | | | | | | | |
| *Morone americana* | Quality | **< 0.001** | **< 0.001** | 0.0074 | **< 0.001** | 0.0408 | **< 0.001** | 0.2079 | 0.8923 |
| | | **< 0.001** | **< 0.001** | | **< 0.001** | 0.0339 | **< 0.001** | | 0.9260 |
| | Quantity | **< 0.001** | 0.0452 | 0.0743 | 0.0220 | **< 0.001** | **0.0026** | **< 0.001** | 0.7179 |
| | | **< 0.001** | 0.0505 | | 0.043 | | | | |
| **Crab:** | | | | | | | | | |
| *Callinectes sapidus* | Quality | **< 0.001** | **0.0022** | **0.0010** | **0.0014** | 0.0197 | **< 0.001** | 0.0607 | 0.5840 |
| | | | | | | 0.0081 | **< 0.001** | | 0.5550 |
| | Quantity | 0.6902 | 0.0099 | 0.5916 | **< 0.00191** | 0.7545 | **< 0.001** | 0.0129 | 0.0568 |
| | | 0.7591 | 0.0135 | | **0.0015** | 0.4273 | **< 0.001** | | 0.0481 |

Note:
Significant results are shown in boldface font.

therefore, threshold measures of genomic quality will be useful to the field of biodiversity genomics. For now, the 9 kb is a practical standard. Longer electrophoreses times (and bigger gels) and more sophisticated and expensive methods would be required for further separation and discernibility of larger fragment lengths (≥ 23 kb). We use the generic "*X* kb" name for our method, to allow for other size standards, such as the 23 kb *Hin*dIII, for future application and computability. The GGBN data standards were developed with this flexibility in mind.

The method proposed does not depend on the choice of fragment size as a threshold for "genomic quality." One could as well pick the *Hin*dIII 564, 2,027 or other fragment sizes as a standard. From the point of view of a biodiversity tissue and DNA repository, whose samples may have been collected years ago and under difficult field conditions, or whose future samples may require difficult field conditions, we propose that 9 kb is, given current technology, a pragmatic value. Whatever the standard chosen, the threshold percentage of gDNA also implies that the extraction will contain fragments much larger than the actual threshold value. Importantly, the "DNA Threshold" and the "Percent above Threshold" standards in the GGBN Gel Image Vocabulary of the GGBN Data Standard (http://terms.tdwg.org/wiki/GGBN_Data_Standard) provide a computable number for comparative values. These values coupled with the gel images allow the researcher to reach their own conclusions on the quality of gDNA for their specific needs. Various user communities can establish discrete bins using these computable statistics such as the four star method proposed for vertebrate genomics (*Wong et al., 2012*).

Of course, genomic DNA of many small organisms, such as certain arthropods, nematodes, meiofauna, and other microscopic organisms is generally difficult to

visualize on agarose gels, yet suitable amounts of genomic sequence data can be successfully amplified from such organisms (e.g. *Blaimer et al., 2015*). We also realize that "degraded" gDNA, < 9 kb for example, can still be used for myriad analyses (e.g. sequence capture, ultra-conserved elements, etc.), including complete genome sequencing, such as the Neanderthal genome (*Prüfer et al., 2014*). Indeed, ancient DNA rarely exceeds 100 bp. However, as biodiversity scientists seek to preserve samples from all major clades of the tree of life, from all biomes, practical and economical field techniques must be developed, and in turn, the effectiveness of such techniques should be quantitative.

Whole genome sequencing will advance technically to use very long fragment sizes, as longer reads provide higher quality assemblies (*Schatz, Delcher & Salzberg, 2010*). Therefore, for plants and animals that can easily be visualized on an agarose gel, we recommend the "greater than *X* kb" method as a standard for biodiversity biobanking laboratories to report the quality of gDNA extracts.

Typically, most library preparation methods to date begin with shearing gDNA to sizes compatible with the maximum size range of most HTS platforms (e.g. 300–500 bp). Therefore, one might question why we should be concerned with large pieces of intact gDNA prior to library preparation. Mechanical shearing, or sonication, shear gDNA randomly across the genome, whereas degradation can cause shearing in non-random places, and in the same places repeatedly, possibly leading to biased HTS results (*Zackin & Ge, 2010*; *Choi et al., 2002*). Furthermore, the use of large insert mate-pair libraries up to 25 kb can increase the efficiency of genomic structure analyses (*van Heesch et al., 2013*).

## Preservation methods

In our tests, salt-saturated DMSO/EDTA buffer and DNAzol are better at preserving high-quality (> 9 kb) gDNA than other methods such as direct storage in liquid nitrogen (−190 °C) or −20 °C storage (Figs. 2A and 3; Table 3). Saturating tissues with storage buffer immediately is also important, as significant DNA degradation can occur, even within three hours time after death (Figs. 2B, 4 and 5; Table 3). Temperature appears to have less of an effect on tissue preservation for overall size-quality of gDNA (Figs. 8 and 9; Table 3). Therefore, time before preservation and preservation method (buffer vs. frozen), and interactions between these factors, have the biggest influence on gDNA quality when measured as size, for both the fish and crab tissue samples (Table 3).

Currently, many genetic researchers working on non-model organisms are under the impression that directly freezing fresh tissue is the best way to preserve gDNA, and the faster and colder the method of preservation, the better (e.g. *Wong et al., 2012*). Liquid nitrogen can be expensive, and both liquid nitrogen and dry ice can be difficult to obtain and transport in certain countries and under remote field conditions. Our results show that putting tissue directly into buffers, such as the salt-saturated DMSO/EDTA or DNAzol is actually better than directly into liquid nitrogen or −20 °C storage, without any buffer for fish (Fig. 4; Table 1), and putting tissues directly into buffers or liquid

nitrogen alone is far better than −20 °C storage for the crab (Fig. 5); albeit all of our samples were flash-frozen first in liquid nitrogen without any buffers prior to treatment.

If further research corroborates these result, this is good news for molecular biologists collecting field samples of genomic material. Salt-saturated DMSO/EDTA is easy and inexpensive to make in the lab and is more easily transported than reagents such as 95% EtOH. We suspect, based on our observations, that the best method of preservation is to allow fresh tissue material to soak in the salt-saturated DMSO/EDTA buffer for approximately 1 h (depending on amount of tissue) and then preserving it in liquid nitrogen, or −20 °C for transportation and/or long term storage.

Interestingly, 95% EtOH appears to be just as good as DMSO for preserving DNA quality in crab tissue (Fig. 5), but not so for fish (Fig. 4). Ethanol can cause extensive, crude dehydration of animal tissues, which may cause fragmentation of gDNA (*Gaither et al., 2011*). Some research supports that ethanol performs better in invertebrate tissues (*Williams, 2007*), perhaps in insects because it can more easily penetrate the cuticle and exoskeletons. Our crab tissue was removed from the shell prior to preservation, which may have improved the performance of EtOH in our study. Regardless, the DMSO/EDTA buffer and 95% EtOH each performed better than direct cryopreservation for the fish and crab tissues, respectively.

Challenges of sufficient amounts of total DNA extracted and concentration (ng DNA/mg tissue) can be overcome by increasing the amount of starting material, and/or combining extractions from several separate extractions of the same starting material source. Here, we have shown that time since death is the biggest factor in gDNA concentration for the fish, but this does not seem to be a factor for the crab tissue (Table 3). Temperature is important for the preservation of fish tissue in terms of quantity (Table 3).

Our study was limited to two organisms, one vertebrate, the white perch (*Morone americana*) and one marine invertebrate, the blue crab (*Callinectes sapidus*). Given the variable results, it is hard to generalize a standard approach. Instead, our methods should be tested for a variety of organisms across the tree of life where tissues are not similar. It is known that marine invertebrates in particular show a wide range of variation in quality and quantity of DNA extracted from tissues preserved using different methods (*GIGA Community of Scientists, 2014*; *Gaither et al., 2011*). If biorepositories are to function in their highest capacity, these results should galvanize the community for further testing of genomic preservation for all major groups of life so that collections of the future are done in the most effective manner.

## ACKNOWLEDGEMENTS

All or portions of the laboratory and/or computer work were conducted in and with the support of the L.A.B. facilities of the National Museum of Natural History (NMNH) or its partner labs. Specifically, we thank A. Driskell and N. Agudelo for help with designing and implementing the DNA extraction tests, and L. Weigt, J. Hunt, and M. Kweskin for L.A.B. facilities and resources. We thank C. Baldwin (NMNH) and R. Aguilar (SERC)

for help with obtaining specimens, S. Whittaker for use of highly sensitive balance, and V. Gonzales for comments on the manuscript.

### Funding

This project was funded by the Smithsonian Institution, National Museum of Natural History's Global Genome Initiative and Laboratories of Analytical Biology (L.A.B.). The funders had no role in study design, data collection and analysis, decision to publish, or preparation of the manuscript.

### Grant Disclosures

The following grant information was disclosed by the authors:
Smithsonian Institution, National Museum of Natural History's Global Genome Initiative and Laboratories of Analytical Biology (L.A.B.).

### Competing Interests

Sean G. Brady is an Academic Editor for PeerJ.

### Author Contributions

- Daniel G. Mulcahy performed the experiments, analyzed the data, wrote the paper, prepared figures and/or tables, reviewed drafts of the paper.
- Kenneth S. Macdonald III conceived and designed the experiments, performed the experiments, analyzed the data, wrote the paper, prepared figures and/or tables, reviewed drafts of the paper.
- Seán G. Brady conceived and designed the experiments, analyzed the data, wrote the paper, prepared figures and/or tables, reviewed drafts of the paper.
- Christopher Meyer conceived and designed the experiments, performed the experiments, analyzed the data, wrote the paper, prepared figures and/or tables, reviewed drafts of the paper.
- Katharine B. Barker wrote the paper, prepared figures and/or tables, reviewed drafts of the paper.
- Jonathan Coddington conceived and designed the experiments, wrote the paper, prepared figures and/or tables, reviewed drafts of the paper.

### Animal Ethics

The following information was supplied relating to ethical approvals (i.e., approving body and any reference numbers):
    Smithsonian Institution, Animal Care and Use Committee (ACUC).

### Data Deposition

    The data is included in the manuscript.

## Supplemental Information

Supplemental information for this article can be found online at http://dx.doi.org/10.7717/peerj.2528#supplemental-information.

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
