# Peer review of "Greater than X kb: a quantitative assessment of preservation conditions on genomic DNA quality, and a proposed standard for genome-quality DNA"

_PeerJ, doi:10.7717/peerj.2528_

## Round 0.1 · original submission · Minor Revisions

Overall both referees are supportive of the work and the value of the recommendation to the field. However, both also have significant suggestions for improvement of the text and some reservations (that I must say I share) about generalization of the results from these two taxa to all species. There are multiple examples (e.g., Dawson et al. 1998 Molecular Marine Biology and Biotechnology 7:145-152; Gaither et al. 2011 Coral Reefs 30:329-333) showing wide variation among marine invertebrate taxa (particularly sponges, tunicates, cnidarians) in preservation methods. I agree with the referees that making strong general recommendations without inclusion of any of these common "problematic taxa" seems a little premature - there should at least be some mention of known difficulties with some marine invertebrate taxa and appropriate caveats on the generalization of recommendations for all taxa.

On the other hand, it seems important to make such recommendations, and I agree with the authors that we should have something in the literature to set a bar for future work. The reviews are split, but the recommendations appear to me to be relatively straightforward and should be possible to deal with in revision. Thus, I am returning a decision of minor revisions and if the referees are satisfied with the revisions, I expect the paper will be acceptable for publication.

·

Basic reporting

This is a review for the PeerJ article “Greater than X kb: A quantitative assessment of preservation conditions on genomic DNA quality, and a proposed standard
for genome-quality DNA” by Daniel G. Mulcahy.,et al . (#11446)

This manuscript meets the basic formatting requirements of PeerJ. The content is self-contained and original. All raw and supplementary data have been made available.

Most of the Figures are relevant and clearly described. The graphed results of Figs 4-5, 8-9 are interesting, but there is not enough differentiation when symbols overlap due to the black color of each. Is there a way to use colored symbols, or intersperse white symbols, to better distinguish those that overlap? This would make interpretations much easier.

Experimental design

In this study, a new approach to assess the quality of animal genomic (gDNA) is described. The authors describe a new quantitative measure for gDNA based on size of fragments after gel electrophoresis and digital capture with ImageJ software.
The four questions posed in this study are highly relevant and practical : 1. Can DNA quality (in terms of fragment length) be measured quickly, consistently, and economically? 2. How does preservation method (buffers vs. temperature) affect DNA quality? 3. How does time since death affect DNA quality? 4. How does storage temperature (in various buffers) affect DNA quality?

This paper actually has two major themes - a) assessing DNA quality by fragment size quantitation and b) assessing the best DNA preservation methods to obtain (a).

The problem of evaluating gDNA quality is sufficiently described as an issue for the current age of “high-throughput” DNA sequencing, where quality is important. I agree that this is important research, and the parameters for assessing and deriving optimal DNA should be rigorously and empirically determined. The goals of the manuscript are highly commendable.

The authors assess “seven tissue preservation methods, time since death, preservation method (i.e. buffers vs. cold temperatures), and storage temperature of various buffers over time.” The experimental design and methods to evaluate these variables appear sound and performed using high standards for quality and accuracy.

Validity of the findings

The authors have addressed most of the primary questions posed above, which makes this is a very commendable manuscript.

The ImageJ approach to assessing gDNA quality is valid. original and quantitative.

For the DNA preservation methods, a main concern for the results is that with so many variables (e.g. ~ 5 preservation buffers, 3 different storage temperatures) to be tested, that this project’s empirical approaches may be overly ambitious. Yet on the other hand, testing was done on only two species (one vertebrate and one invertebrate), and so overall applicability across a wide taxonomic spectrum becomes less plausible. This point was carefully acknowledged by the authors but should be reconciled with the main concern above.

Use of ANCOVA results for the Time and Temperature experiments were very important and should be lauded, as one way to statistically assess which method may be optimal.

In spite of the above there were still several problematic areas that should be addressed by the authors:

• The choice of “9 kb” as fundamental quality and size metric on Line 396 – “…. 9 kb is, given current technology, a pragmatic value.”, is not entirely clear and appears arbitrary. More explanation should be given, since if single molecule sequencing (e.g. Nanopore minion) improves, 9 kb contigs could actually be on the smaller side. Also the highest Lambda bands resolvable on standard agarose gels is the 23Kb.
Can the authors point to more support for an industry or community standard? Also the Loman and Quinlan, 2014 citation points to a nanopore-based toolkit and technology, which has still not been widely used or accepted even at this time. Thus caution should be exercised when citing this technology.

• Because the terms “DNA Threshold” and “Percent above 28 Threshold”, are important metrics in this paper, they should be better and more deeply defined in the text, preferably in the METHODS and not through a URL at line 399 (Moreover, this reviewer could not reach the website http://terms.tdwg.org/wiki/GGBN_Data_Standard - GGBN_Gel_Image_Vocabulary), which is another reason to include a clear definition and original reference for these metrics. For example, where were these metrics originally defined? These definitions could be incorporated into the paragraph 391-402, which based on the preceding discussion should be re-worded and clarified.

• A discussion of small organism DNA yields, problems of heterozygosity etc could be enhanced in lines 403-406, and could cite the GIGA white paper which describes several of these issues (GIGA-COS 2014. Developing Community Resources to Study Diverse Invertebrates. Journal of Heredity. 105:1-18).

• Although it may require some more thought and elaboration, and because the results for only two species will be difficult to extrapolate broadly to other taxa or species, perhaps the authors should try to devise a broader quantitative scoring system of gDNA quality and preservation methods based on the current results. This could be as simple as the one to four star rating method of Wong et al 2012 (Tissue sampling and standards for vertebrate genomics. GigaScience 1:8.595 DOI: 10.1186/2047-217X-1-8), or developing a more elaborate scoring method (e.g. 1-10; 10 as best) based on the variables described by the authors.

Additional comments

In a related context, the authors should also mention if these methods could be applied to pulsed field gel electrophoresis runs. This is based on the authors’ own emphasis on higher molecular weight fragments as a target, and that technology trends will likely move towards targeting larger (not smaller) fragments for templates. This will add more dimension and consistency to the overall analyses.

Minor suggested changes and typos:

Line 74 – suggest change “For genomic quality DNA, it is preferred to be mostly intact” to

“Genomic quality DNA should preferably be mostly intact”

Line 361 – please remove the “a” in the sentence “We tested whether a single, or multiple researchers could…”

Another reference that the authors may find useful for preservation methods is the grey literature – Deep-sea coral collection protocols. P Etnoyer, SD Cairns, JA Sanchez, JK Reed, JV Lopez, WW Schroeder, ... NOAA Technical Memorandum NMFS-OPR 28.

Reviewer 2 ·

Basic reporting

No Comments

Experimental design

No Comments

Validity of the findings

No Comments

Additional comments

This is a great paper. Its well written and presents data, that may not be particularly sexy in the eyes of some (I'm not one of those), it super useful! I have a few suggestions mostly centred on what I thought was an under emphasis on the tests of the preservation methods.
So the paper has two primary aims
1. Testing a gel method for determining DNA quality (one that would become a field standard)
2. Test the effect of various preservation methods on DNA quality
Reading the abstract and introduction the latter (aim #2) seems to be a secondary topic but I found it to be one of the more useful aspects of the paper. For instance, about 2/3 of the way through the abstract the author say “We also present data” (Line 30) and then proceed to talk about the preservation test methods. I suggest the authors shuffle the abstract and intro a bit to give more equal emphasis to both aspects of the paper. Also logically preservation methods should come first and then the gel method for determine the quality of extractions. This is a less important criticism as it feels like the authors are keen to prioritise aim#1 but logic tells me they should be switched in order.
Also in the abstract the results of the preservation tests are truncated to a single sentence. Did the results differ between the fish and crab? Which buffer was best?
Line 40-43: Most NG library prep methods work better when you start out with high molecular weight DNA so the emphasis on sequence read length seems incomplete and may be driving the emphasis on aim #1 but again aim #2 is arguable more important (you can’t get high molecular wt DNA without proper preservation) and likely to be of broader interests (folks outside GGBN).
Line 113: Where were the study organisms obtained (wild population or cultured)? I’m really just curious. I’m not sure it makes any difference to the study.
Line 123-124: Does this mean that a single fish/crab was used for both time and temp experiments or one fish/crab was used for the time experiment and another fish/crab was used for the temp experiment. Please clarify.
Line 209: Does the imager used matter? Should something be said here about consistency in the way the images are made or does it matter?
Figures 4&5 I found the symbols hard to pull apart in Fig. 5. Since PeerJ is online I assume there is no disadvantage to using color…or slightly offset symbols so the reader can differentiate them. Same issue with Fig. 8.
Figure4: should be “legend” and “sat at room temperature”. After the latter I would include the time samples were held in preservative prior to extraction. I assume the error bars are SD. Please specify in legend.
Figure 5: include “see legend” as in Fig. 4 (or not in any) please check all legends for consistency.
What are the double bands in the later gels (3 onward)? I see this regularly and always assume it’s an artefact.

---

## Round 0.2 · accepted · Accept

I am happy to say that everyone is satisfied with your revisions and I am happy to accept your paper at this point.

·

Basic reporting

Please check consistency of the formatting for the Droege et al, In Press reference (upper vs lower case).

Experimental design

No additional comments

Validity of the findings

No additional comments

Additional comments

The authors have satisfactorily addressed all of my concerns and previous comments. This will be a very good and useful contribution to the genomic sciences.